# A Versatile and Robust Serine Protease Inhibitor Scaffold from *Actinia tenebrosa*

**DOI:** 10.3390/md17120701

**Published:** 2019-12-12

**Authors:** Xingchen Chen, Darren Leahy, Jessica Van Haeften, Perry Hartfield, Peter J. Prentis, Chloé A. van der Burg, Joachim M. Surm, Ana Pavasovic, Bruno Madio, Brett R. Hamilton, Glenn F. King, Eivind A. B. Undheim, Maria Brattsand, Jonathan M. Harris

**Affiliations:** 1Institute of Health and Biomedical Innovation, Queensland University of Technology, Brisbane, QLD 4059, Australia; xingchen.chen@connect.qut.edu.au (X.C.); darren.leahy@hdr.qut.edu.au (D.L.); j.vanhaeften@qut.edu.au (J.V.H.); chloe.vanderburg@connect.qut.edu.au (C.A.v.d.B.); a.pavasovic@qut.edu.au (A.P.); 2School of Biomedical Sciences, Faculty of Health, Queensland University of Technology, Brisbane, QLD 4000, Australia; p.hartfield@qut.edu.au; 3School of Earth, Environmental and Biological Sciences, Science and Engineering Faculty, Queensland University of Technology, Brisbane, QLD 4000, Australia; p.prentis@qut.edu.au; 4Institute for Future Environments, Queensland University of Technology, Brisbane, QLD 4000, Australia; 5Institute for Molecular Bioscience, University of Queensland, St Lucia, QLD 4072, Australia; b.madio@uq.edu.au (B.M.); glenn.king@imb.uq.edu.au (G.F.K.); 6Centre for Advanced Imaging, University of Queensland, St Lucia, QLD 4072, Australia; b.hamilton@uq.edu.au (B.R.H.); eivind.a.b.undheim@ntnu.no (E.A.B.U.); 7Centre for Biodiversity Dynamics, Department of Biology, Norwegian University of Science and Technology, 7491 Trondheim, Norway; 8Centre for Ecological and Evolutionary Synthesis, Department of Biosciences, University of Oslo, PO Box 1066 Blindern, 0316 Oslo, Norway; 9Department of Medical Biosciences, Umeå University, 901 87 Umeå, Sweden; maria.brattsand@umu.se

**Keywords:** *Actinia tenebrosa*, kallikrein-related peptidases, Kunitz inhibitor, serine protease, mass spectrometry imaging, molecular dynamics simulation

## Abstract

Serine proteases play pivotal roles in normal physiology and a spectrum of patho-physiological processes. Accordingly, there is considerable interest in the discovery and design of potent serine protease inhibitors for therapeutic applications. This led to concerted efforts to discover versatile and robust molecular scaffolds for inhibitor design. This investigation is a bioprospecting study that aims to isolate and identify protease inhibitors from the cnidarian *Actinia tenebrosa.* The study isolated two Kunitz-type protease inhibitors with very similar sequences but quite divergent inhibitory potencies when assayed against bovine trypsin, chymostrypsin, and a selection of human sequence-related peptidases. Homology modeling and molecular dynamics simulations of these inhibitors in complex with their targets were carried out and, collectively, these methodologies enabled the definition of a versatile scaffold for inhibitor design. Thermal denaturation studies showed that the inhibitors were remarkably robust. To gain a fine-grained map of the residues responsible for this stability, we conducted in silico alanine scanning and quantified individual residue contributions to the inhibitor’s stability. Sequences of these inhibitors were then used to search for Kunitz homologs in an *A. tenebrosa* transcriptome library, resulting in the discovery of a further 14 related sequences. Consensus analysis of these variants identified a rich molecular diversity of Kunitz domains and expanded the palette of potential residue substitutions for rational inhibitor design using this domain.

## 1. Introduction

Serine proteases are pivotal regulators of multifarious cellular activities through the activation of protein precursors, such as prohormones, pro-enzymes [1], and protease-activated receptors [2]. Serine proteases also play important roles in disease progression [3] and cancer [4]. Accordingly, considerable research interest is focused on isolating and identifying serine protease inhibitors for therapeutic translation. An increasingly common approach to inhibitor design is to bioengineer a naturally occurring molecular scaffold to target a given protease with high selectivity and potency [5,6,7]. An ideal inhibitory scaffold should have the following key qualities: it needs to be amenable to amino-acid substitution, it should be physically robust to enable long-term storage, it needs to possess in vivo stability, and it should be capable of making specific interactions with its target proteases. The bioscaffolding approach has seen many successes with examples including ecallantide (Kalbitor^®^), a variant of tissue factor pathway inhibitor-1 [8], which is now being used clinically for the treatment of hereditary angioedema [9], and hirudin variants, such as lepirudin (Refludan^®^), which is used for treatment of heparin-induced thrombocytopenia [10]. The success of these bioengineered protein-based drugs relies on both the potency and the selectivity of inhibition they show for their targets. The ability to selectively target a given protease is especially important for therapeutics as exemplified by the failure of early metalloprotease inhibitors undergoing development for use in anti-cancer therapies [11]. Off-target inhibition of the related ADAMs (a disintegrin and metalloproteinases) proteases by these inhibitors caused the failure of clinical trials and resulted in the current quest for highly specific protease inhibitors [11]. 

The kallikrein-related peptidases (KLKs) are a family of serine proteases which share considerable structural homology. Hence, like the metalloproteases and ADAMs proteases, their targeted inhibition is complicated by a potential for off-target activity. Notwithstanding these challenges, inhibition of KLKs is of interest because of their roles in normal physiology and disease [12]. To date, KLKs have been strongly linked with prostate cancer [13], skin desquamation [14,15], skin disorders [16], and neurodegeneration [17]. Despite having overlapping substrate specificities, it is apparent that KLKs often act in concert to produce cascades of proteolytic activities similar to those seen in blood clotting and apoptosis [18,19]. In turn, this has led to a growing interest in these enzymes as important potential points of therapeutic intervention [5,20] and an associated drive to design KLK-selective inhibitors.

Previous work in the area of venomics suggested that sea anemones might be a good source of protease inhibitors (see References [21,22] for a comprehensive review). Furthermore, the sea anemone’s ecological niche is challenging, with the organism facing diurnal atmospheric exposure, solar irradiation, and dehydration without the benefit of a hardy and protective cuticle. Accordingly, a sea anemone proteome might be expected to possess an intrinsically robust structure reflecting its uncompromising ecological niche. The first protease inhibitor from sea anemones was identified in 1972 in *Anemonia sulcate* [23]. Since then, protease inhibitors have been isolated and characterized from several species of sea anemones. Shiomi et al. [24] isolated four inhibitor variants from *Actinia equina* termed *Actinia equina* Protease Inhibitor 1-4 (AEPI-I, II, III, and IV). AEPI-I to IV displayed potent inhibition of trypsin and moderate inhibition for chymotrypsin, plasmin, and plasma kallikrein, although *K*_i_ values were not determined [24]. More recently, García-Fernández et al. successfully produced a highly specific elastase inhibitor by engineering the *Stichodactyla helianthus* protease inhibitor (ShPI-1) [25], demonstrating the biotechnological potential of sea anemone protease inhibitors.

In the present study, two protease inhibitors were isolated from *Actinia tenebrosa*, the most common species of sea anemone distributed along the eastern and southern coast of Australia. Their amino-acid sequences were elucidated, and a Kunitz inhibitory domain was identified in both inhibitors. Their inhibitory potencies against trypsin, chymotrypsin, KLK5, KLK7, and KLK14 were assessed. To understand their differential inhibition of KLKs, homology modeling and molecular dynamics simulations were undertaken, which revealed ideal qualities for inhibitor engineering. Furthermore, an *A. tenebrosa* transcriptome library was searched for Kunitz domain homologs, yielding another 14 sequences. These sequences were subjected to consensus analysis, providing fresh substitutional avenues for Kunitz scaffold-based inhibitor design.

## 2. Results

### 2.1. Isolation of Two Low-Molecular-Mass Protease Inhibitors from A. tenebrosa

Two low-molecular-mass protease inhibitors, which were termed ATPI-I and ATPI-II, were isolated from total lysates of *A. tenebrosa*. After initial water extraction, gel filtration, and cation exchange HPLC, two discrete fractions with trypsin inhibitory activity were resolved. Each fraction was individually purified by reverse-phase (RP) HPLC to produce the two distinct inhibitors (Figure 1). Homogeneity and molecular masses of these inhibitors were characterized by mass spectrometry analysis on a SELDI protein-chip system, showing average masses of 6719.9 Da for ATPI-I and 6606.5 Da for ATPI-II (Figure 2). 

### 2.2. Sequence Analysis Reveals ATPI-I and ATPI-II Inhibitors Belong to the Kunitz-Type Family

To determine the amino-acid sequence of these inhibitors, ATPI-I and ATPI-II were digested with trypsin, and then analyzed using liquid chromatography/tandem mass spectrometry (LC–MS/MS). Trypsin digest products were resolved by RP-HPLC, ionized, and subjected to a full survey scan. The three most abundant precursor ions identified were selectively fragmented, and the resulting ions (product ions) were subjected to MS/MS scans. For preliminary sequence analysis, the MS/MS spectra were used to deduce the precursor sequences based on the intervals between neighboring ion peaks. Figure 3 shows representative MS/MS spectra and deduced peptide sequences of ATPI-I (Figure 3A) and ATPI-II (Figure 3B). These sequences were then used in searches within the transcriptomic library of *A. tenebrosa* [26], which returned a candidate sequence for each inhibitor. All digested peptide sequences were obtained for both inhibitors by comparing candidate sequences with MS/MS spectra. The calculated and observed *m*/*z* for precursor sequences are summarized in Table 1, which details the full sequences for the inhibitors. The calculated average masses of full-length inhibitors were 6719.43 Da (ATPI-I) and 6604.39 Da (ATPI-II). These masses corresponded well with the experimental average masses obtained by MALDI/MS-TOF, as shown in Figure 2. Both sequences belong to the Kunitz-type inhibitor family with six conserved cysteine residues. BLAST searches revealed that ATPI-I has a similar sequence to AEPI-I (an inhibitor previously isolated from *A. equina*) [24], with an additional alanine at the C-terminus. ATPI-II has an identical N-terminal sequence to AEPI-III (MW 6200 Da) [24]. However since AEPI-III was not fully sequenced [24], it is not currently possible to make comments about the overall level of sequence similarity between ATPI-II and AEPI-III.

### 2.3. ATPI-I Is Localized to the Mesenteric Tissues of A. tenebrosa

In order to gain insight into the potential biological role of these inhibitors, the tissue distribution of ATPI-I messenger RNA (mRNA) was determined by tissue-specific transcriptome analysis. Total mRNA was extracted, sequenced, and quantitated across different tissues, including tentacles, mesenterial filaments, and acrorhagi (Figure 4A,B) [26]. Similar to other cnidarians, sea anemones use tentacles primarily to capture prey and repel predators. whereas mesenterial filaments actively secrete protein components which are chiefly involved in digestion, but may also be secreted for external digestion, prey capture, and defense [30,31]. Acrorhagi are specialized offensive organs involved in intra-species competition, and they have a restricted distribution amongst sea anemones, including *A. tenebrosa* and the closely related species *A. equina* [32]. In *A. tenebrosa*, it is apparent that the ATPI-I transcript is expressed most highly in the mesenterial filaments, with relatively lower levels (3–4 fold) of expression in the acrorhagi (Figure 4B). Expression levels in tentacles are detectable, but markedly low (Figure 4B). Localization of ATPI-I mRNA was consistent with protein distribution observed by MALDI mass spectrometry imaging (Figure 4C,D), suggesting that ATPI-I is most likely involved in regulating functions and processes within the digestive system of *A. tenebrosa*.

### 2.4. ATPI-I and ATPI-II Have Distinctive Inhibitory Profiles toward the Closely Related Proteases KLK5, KLK7, and KLK14 

Kinetic assays were performed to determine inhibition constants (*K*_i_) for both inhibitors toward trypsin, chymotrypsin, and the three major kallikrein-related peptidases in the epidermis, KLK5, KLK7, and KLK14, using corresponding tetrapeptide *para-*nitroaniline (pNA) substrates (Table 2 and Appendix A, Appendix A). ATPI-I and ATPI-II have both tryptic and chymotryptic inhibitory activities. Both peptides strongly inhibited trypsin, with *K*_i_ values of 0.05 nM (ATPI-I) and 0.08 nM (ATPI-II). APTI-I and APTI-II are also potent inhibitors of chymotrypsin, with *K*_i_ values in the nanomolar range (Table 2). These potencies are comparable to those seen with ShPI, a previously reported Kunitz inhibitor isolated from *S. helianthus*, which inhibited trypsin with *K*_i_ ~0.1 nM and chymotrypsin with *K*_i_ ~1 nM [28]. For the KLKs, ATPI-I exhibited higher levels of inhibitory activity than ATPI-II. ATPI-I showed comparable levels of inhibition against KLK5 (*K*_i_ = 1.6 nM) and KLK7 (*K*_i_ = 1.4 nM), while inhibition of KLK14 (*K*_i_ = 13 nM) was ~10-fold lower. ATPI-II was more potent against KLK7 (*K*_i_ = 2.6 nM) than KLK5 (*K*_i_ = 21 nM) or KLK14 (*K*_i_ = 94 nM). Interestingly, KLK5 and KLK14 are trypsin-like enzymes with a P1 preference for Arg and Lys, whilst KLK7 is a chymotrypsin-like enzyme with a P1 preference for large hydrophobic side chains of Tyr, Phe, and Trp. However, both ATPI-I and ATPI-II possess a basic P1 residue, but inhibit KLK7 more potently than KLK5 or KLK14.

In order to evaluate resistance to heat-induced denaturation, ATPI-I was diluted in pre-heated assay buffer and incubated at 95 °C for 5 min before being assessed in a trypsin inhibitory assay. ATPI-I showed no loss of inhibitory activity after incubation, indicating an ability to withstand elevated temperature and resistance to heat-induced denaturation (Figure 5).

### 2.5. Hydrogen-Bond Analysis Reveals Specific Interactions of ATPIs and KLKs

Despite the high level of similarity between the two inhibitors, they exhibit different inhibition profiles against the KLKs under investigation. To understand the structural basis of this selectivity and affinity, molecular models of ATPI-I and ATPI-II were constructed. Their solution dynamics simulations (20 ns) in unbound form and in complex with KLK5, KLK7, or KLK14 were performed. For all systems, root mean square deviation (RMSD) values of Cα atoms stabilized after 15 ns (Appendix A, Appendix A). Simulation trajectories of the last 5 ns (5000 frames each trajectory) were extracted for further analysis. In the following analyses, suffixes (I) and (II) were used to distinguish residues belonging to ATPI-I and ATPI-II. In unbound form, both inhibitors show increased mobility at the canonical loop regions (residues 13–18) and regions lacking well-defined secondary structure (residues 26–28 and 39–40), as reflected by the root-mean-square fluctuation (RMSF) of Cα atoms (Figure 6). Overall, the complexed inhibitors were less dynamic compared to the unbound forms, where the increased stability in the primary contact loop is most noticeable. In addition, a decrease in Cα RMSF for residues 39–40 was observed, suggesting a secondary contact region.

Further analysis of hydrogen bonds (Figure 7) and close contacts (distance < 4 Å, Appendix A, Appendix A) revealed the key residues involved in inhibitor–protease interactions. The primary contact loop corresponds to P4–P3’ subsites with the P1 residue Arg16 (I) or Lys16 (II) at the center point. The secondary contact region is formed by an identical loop, _35_IYGGCG_40_, in both inhibitors. This is consistent with the observed Cα RMSF of free and complexed inhibitors, which show that the solvent-exposed canonical loop and 35–40 loop is stabilized upon complex formation. A weak contact site was observed at the third loop, with residue Glu47 (I, II) in proximity to the “60-loop” of KLKs. 

The number of H-bonds in the interface reflected the divergent inhibitory activities of the two inhibitors. In the KLK5–inhibitor complexes, ATPI-I engages an average of 8.2 intermolecular H-bonds largely confined to the P3–P1 residues (Figure 7). The side chain of Arg16 (I) is inserted into the S1 pocket framed by residues 189–195 and 213–218 and forms three strong H-bonds (occupancy > 85%) with Asp198–Ser190. The backbone atoms of ATPI-I, including Arg16 (I) O, Arg16 (I) N, Cys15 (I) O, and Lys14 (I) O, also form stable H-bonds with KLK5. The number of contacts is reduced in the interface of the KLK5–ATPI-II complex, especially the S1 pocket (Figure 7A,B). An additional weak contact site was identified at residue Glu47 (I, II) which forms an H-bond with KLK5 residue Lys60, which does not exist in other complexes.

The KLK7 S1 pocket has a hydrophobic environment bordered by residues Asn189 and Ala190 at the bottom of the pocket. Arg16 (I) and Lys16 (II) are able to fit into the KLK7 S1 pocket and interact with the backbone atoms of Ala190 and Thr217 (Figure 7C,D). At the interface of the KLK7–inhibitor complexes, the inhibitor forms a set of intermolecular H-bonds mediated by backbone atoms extending from P3 to P2’. The only highly favored side-chain interactions involved are the guanidino group of the P3 Arg14 of ATPI-II and His99 of KLK7 (occupancy > 95%). Both peptides are nanomolar inhibitors of KLK7, suggesting a more important role for backbone interactions than specific side-chain interactions.

The kinetic assays indicated that inhibition of KLK14 by ATPI-I and ATPI-II has the lowest *K*_i_ values among the three KLKs. Structural overlays show that the side chains of the P3 residues Lys14 (I) and Arg14 (II) were deflected away from the interface and did not appear to interact with bound KLK14 (Figure 7E,F). Consistent with this observation, the S3 subsite of KLK14 is framed by large side chains of Trp215, Met217, and His99, suggesting a binding preference for small hydrophobic residues. This is more conspicuous in the KLK14–ATPI-II complex, since the presence of Arg97 in KLK14 potentially constrains access of the longer sidechain of Arg14 (II), whilst this position is occupied by the smaller side chains of Pro97 in KLK5 and Gln97 in KLK7.

### 2.6. Sequence Diversity of Kunitz Domains in A. tenebrosa Inhibitors Indicates Functional Diversity

To explore the potential sequence diversity of Kunitz-type inhibitors in *A. tenebrosa*, a transcriptomic library from the organism was searched with the consequent discovery of 14 candidate sequences possessing Kunitz domains. Following on from the isolation and characterization of ATPI-I and ATPI-II in the current study, the additional 12 identified sequences were given the designations AT03–AT14 (Figure 8A). Predictive molecular models of AT03–AT14 were generated by homology modeling using the YASARA suite of programs [34]. The sequences have 42%–95% identity to ATPI-I and, as expected, the modeled structures have similar Cα trace conformations to ATPI-I, with increased flexibility at both termini. Conservation scores at each residue position were calculated based on the residue similarity in the multiple alignment. The majority of residues cluster at opposite ends of the conservation score bar; 27 residues have a conservation score from 0–3 (more variable), and 21 residues have a score of 9–11 (more conserved) (Figure 8B). 

In addition to the six conserved Cys residues, Leu8, Gly13, Tyr36, Gly37, Gly38, Gly41, Asn42, Asn44, Asn45, and Phe46 are also completely conserved (conservation score 11). At positions 17, 23, 24, 36, and 48, substitutions are restricted to residue side chains with similar properties, such as Ala/Gly, Tyr/Phe, and Thr/Asn (conservation score 9–10). These highly conserved residues are either hydrophobic or polar uncharged residues comprising the two hydrophobic cores of the Kunitz domain, which are likely to play an important role in stabilizing the Kunitz scaffold. 

In the primary loop, P1, P3, and P2’ positions show considerable diversity. At P3, Lys or Arg occurs more frequently than other residues. Proline is the typical P3 residue in mammalian Kunitz inhibitors; however, proline only appears in one identified sequence (AT10). There are six different residues occupying the P1 residue space: Arg, Lys, Met, Gly, Tyr, and His, whereas Arg and Lys appear in nine out of 14 sequences. Preponderance of Arg and Lys probably reflects natural selection against trypsin-like proteases. However, the typical P4–P1 combinations of GPCK or GPCR found in mammalian Kunitz-type trypsin inhibitors are not present in the identified sequences in *A. tenebrosa*. P1-Met is usually associated with inhibition of chymotrypsin or elastase. 

Compared to the primary loop, the secondary contact site is more conserved. Variations only occur at residues 35 and 40. At residue 35, Ile appears most frequently, occupying this position in eight out of 14 sequences (57%), followed by Phe (21%). The other residues that appear at position 35 are Tyr, Asn, and His, which each appear once. Glycine is the most prevalent residue at position 40, while Arg, Ser, Gln, or Leu appear as single substitutions. The third contact site revealed by molecular dynamics simulations is located at residue Glu45 of ATPI-I and ATPI-II. Interestingly, they are the only two sequences possessing a Glu at this position. Other substitutions in the AT03–AT14 sequences at residue 45 include Arg (36%), Ala (29%), Lys (7%), and Asp (7%) (Figure 8A).

Residues are more conserved at the stable hydrophobic cores of the Kunitz domains. When the conservation levels decreased, residues moved toward both ends of the domain, especially in the primary loop, the β-turn (residues 26–29), and the C-terminal α-helix (Figure 8C). These are among the most flexible regions seen in molecular dynamics simulations of ATPI-I and ATPI-II (Figure 6), suggesting a link between conservation and structural stability. In order to estimate the per-residue contribution to molecule stability, *i*n silico alanine scanning mutagenesis was performed by calculation of the change in free energy (ΔΔG) after mutating each of the residues to alanine, and it was averaged over all 14 sequences. This resulted in ΔΔG values ranging from −0.2 kcal/mol to 5.4 kcal/mol, with an average of 1.6 kcal/mol. All residues were then classified into groups according to their conservation score as shown in Figure 9: (A) 9–11 (highly conserved); (B) 5–8; (C) 2–4; (D) 0–1 (most variable). Residues in each group were then compared with their contribution to protein stability evaluated by averaged ΔΔG (Figure 9E).

The most conserved groups (conservation score 9–11) mainly consist of disulfide-bonding cysteine residues and hydrophobic residues that comprise the two hydrophobic cores of the Kunitz domain (Figure 9A). It was not surprising to find that alanine substitution at the cysteine residues (which disrupts disulfide bonds) resulted in large ΔΔG (>3.5 kcal/mol). At the P4 position of the inhibitory loop, a G14A mutation resulted in ΔΔG > 5 kcal/mol. Most of the other residues with a conservation score of 9–11 show a ΔΔG > 2 kcal/mol after mutation to alanine, indicating that substitutions at these positions may destabilize the Kunitz scaffold, and further suggesting their essential role in maintenance of the stability of the scaffold. Mutations at these positions may destabilize the scaffold. Exceptions are Leu8, Asn42, Asn44, Asn45, and Thr/Asn48 located on the surface (Leu8) or in the area between the two cores, which appear more tolerant to alanine mutation. 

At a conservation score of 5–8, residues are located near the hydrophobic core but are solvent-exposed, and mutations of charged residues begin to emerge (Figure 9B). For residue positions with conservation scores 2–4, alanine scanning yielded an average ΔΔG of ~1–2 kcal/mol, indicating a superior tolerance for mutations. Charged residues were frequently observed on solvent-exposed surfaces (Figure 9C), suggesting a role in molecular recognition, binding, and solubility. It is noteworthy that Arg/Lys residues at position 27 and 29 located at the β-turn are thought to be key residues for interaction with potassium channels [35,36]. Thus, AT12 and AT13, which possess Arg27–Lys29, may potentially have a secondary function as ion channel inhibitors.

The most variable amino acids (conservation score 0–1) are all solvent-exposed residues mainly gathered around the β-turn, N-terminus, and C-terminus, opposite the inhibitory loop, with the exception of residues 10 and 45 located in the middle part of the domain. Substitutions of all types exist at these positions, including charged residues. Alanine scanning showed that replacing these residues results in ΔΔG < 1 kcal/mol.

Overall, conservation analysis and in silico alanine scanning mutagenesis yielded similar results. Linear regression analysis indicates a moderate positive correlation (*r* = 0.73) between residue conservation score and contribution to stability (Figure 9E). In other words, the most invariable residues are more important for protein stability. Residues with important functional roles are highlighted by conservation scores of 2–4. Although sequence analysis does not reveal the functional roles of these substitutions, the data still provide important indications of tolerance to substitution at these positions.

This analysis demonstrates considerable sequence diversity amongst the Kunitz homologs present in *A. tenebrosa*. Overall, this is a valuable resource as the molecular diversity occurs in the context of a single organism, rather than an alignment of sequences from many different organisms as is frequently the case when using sequence diversity for molecular design purposes. In turn, the observed diversity probably reflects a response to a single set of environmental conditions. Furthermore, the contact sites that bind to proteases showed considerable diversity, which may drive functional specificity, whilst the conserved regions are likely to be critical for molecular stability.

## 3. Discussion

Introducing molecular diversity into a naturally occurring inhibitory framework is an established technique to generate new variants that can engage a selected target [37,38,39]. However, the substitutions accompanying this diversification can have deleterious effects on the integrity of the scaffold. We previously reported on this phenomenon when carrying out substitutions within the sunflower trypsin inhibitor (SFTI-1) [6], a robust, 14-residue cyclized peptide stabilized by a bisecting disulfide bond and extensive internal H-bonds. We demonstrated that substitutions within the SFTI-1 binding loop could redirect its inhibitory selectivity and potency [6]. However, these substitutions disrupted the inhibitor’s H-bond network, leading to instability and reduced inhibitory potency. In turn, this required further engineering to restore H-bonding [14,40,41]. Although SFTI-based variants proved to be both potent and selective inhibitors of serine proteases, their small size and hydrophilic nature mean that they are rapidly removed from systemic circulation [41], which was one of the factors driving our investigations in the present study. 

Sea anemones are a rich source of bioactive proteins and peptides, including neurotoxins, ion channel blockers, and protease inhibitors [22,42]. In the current study, two protease inhibitors (ATPI-I and ATPI-II) were isolated from *A. tenebrosa* and comprehensively characterized. The sequences of both inhibitors were fully determined, showing that both variants belong to the Kunitz inhibitor family. Furthermore, both ATPI-I and ATPI-II were shown to inhibit three critical epidermal kallikreins, KLK5, KLK7, and KLK14. Despite the high sequence similarity (85%) between ATPI-I and ATPI-II, considerable differences in their inhibitory potencies for selected proteases were observed. It is noteworthy that ATPI-I showed a 13-fold higher inhibitory potency (in terms of *K*_i_) against KLK7 than ATPI-II. This result is surprising given that the canonical loop sequences of ATPI-I (P4–P2’: GKCRGY) and ATPI-II (P4–P2’: GRCKGY) are remarkably similar; the only differences are substitutions between the basic residues arginine and lysine at the P3 and P1 positions. However, this finding paralleled previous engineering studies with SFTI, where a single amino-acid substitution could completely redirect the inhibitory selectivity of a variant [40,41]. Overall, it is conjectured that ATPI-I and ATPI-II will be promising candidates for engineering highly specific KLK5 and KLK7 inhibitors.

To further understand the role of individual side chains in protease binding, molecular dynamics simulations revealed differential preferences for Arg and Lys in the interface between ATPIs and different KLKs. Although Arg and Lys are both basic residues, Arg contains a guanidinium group that enables it to form more H-bonds and salt bridges than Lys [43]. At the P1 position, Arg shows a broader range of inhibition against the selected KLK proteases in this study. Molecular dynamics simulations suggest that the direct H-bonds between S1 and P1 Arg are the major driver of high affinity in the ATPI-I/protease complexes, while these interactions are decreased when the P1 Arg is replaced with Lys in ATPI-II. This observation is consistent with previous studies showing that KLK5, KLK7, and KLK14 exhibit a preference for Arg over Lys at P1 [44,45]. Importantly, ATPI-I and ATPI-II are able to discriminate between closely related proteases, KLK5 and KLK14, through specific P3–S3 interactions. This feature provides a promising starting point for further inhibitor design using the Kunitz domain scaffold.

Discovery of additional variants through transcriptomics [26] provides important information about the diversity of Kunitz domain sequences in *A. tenebrosa*. The current study conducted a comprehensive functional and structural survey of the complement of Kunitz inhibitors in *A. tenebrosa*. A whole-body transcriptome of *A. tenebrosa* was searched for Kunitz homologs. Sequence analysis of the resulting 14 putative Kunitz domains showed a positive correlation between the degree of residue conservation and the residue’s contribution to molecular stability. Moreover, these Kunitz-type domains displayed considerable diversity in their inhibitory loops, indicating a robust scaffold amenable to additional modifications. Interestingly, residues related to functional diversity emerged at the same level of conservation. Taken together, it could be suggested that these variants originated from the same ancestral gene via gene duplication and later diversified for different biological functionalities [46]. Consistent with this supposition, tissue-specific transcriptomic analysis of ATPI-I revealed that it is most highly expressed in the mesenteric tissue, suggesting a role in regulating functions in the sea anemone’s digestive system, rather than functions associated with prey capture or defense. 

Protein evolution is attracting increasing levels of interest in the field of protein engineering. A prime example is the directed evolution of protein mutagenesis—a laboratory approach which mimics the natural processes to evolve proteins and amplify certain properties. Directed evolution is initiated by expression of a gene library in an organism (typically *Escherichia coli*) under selective pressure, which is followed by iterative screening and selection [47]. A notable success using this method is the development of green fluorescent protein (GFP) variants that display significantly improved fluorescence after iterative rounds of random mutagenesis and selection [48]. However, one of the downsides of directed evolution is that this method usually results in accumulation of large numbers of potential hazardous or neutral mutations. In contrast, natural selection retains only a small fraction of possible sequences that are structurally stable and biologically relevant. Residue conservation defined by natural selection largely restricts the number of positions to be explored for further engineering. In this study, a relationship between structural stability and the degree of residue conservation was demonstrated by in silico alanine scanning mutagenesis. These findings reveal a rich diversity of sequences and expand the palette of potential residue substitutions for rational inhibitor design using this domain. 

## 4. Materials and Methods

### 4.1. A. tenebrosa Collection 

Sea anemones *A. tenebrosa* were collected from rocks at low tide around the base of Point Cartwright in southeast Queensland. Identification was performed using the description available at the University of Queensland’s image collection “invertebrates of the coral sea” (www.gbri.org.au). Once identified, specimens were detached by inserting a spatula under their pedal disc and then gently levered away from their substrate before freezing on dry ice for transport and longer-term storage at −20 °C until processed. Because sea anemones have a simple nervous system, use of *A. tenebrosa* does not require animal ethics approval.

### 4.2. Inhibitor Purification

Whole specimens of *A. tenebrosa* were homogenized by grinding under liquid nitrogen, and the resulting powder was suspended in three volumes of distilled water. The resulting suspension was clarified by centrifugation at 15,000× *g* for 15 min, and proteins in the supernatant were precipitated with 80% ice-cold acetone and pelleted by centrifugation, before being re-solubilized in 0.1 M ammonium bicarbonate. Insoluble material was removed by centrifugation (15,000× *g*, 15 min). The supernatant was then subjected to gel filtration on Sephadex S-200 resin equilibrated in 0.1 M ammonium bicarbonate and eluted with the same buffer at a flow rate of 4 mL/min. Fractions showing protease inhibitory activity were pooled, partially purified using a C18 solid-phase extraction cartridge (Grace, Columbia, MD, USA), lyophilized, and re-solubilized in 25 mM potassium phosphate (pH 6.0) before being applied onto a UNOsphere-S cation-exchange chromatography column. The column was eluted using a linear gradient of 0.0–1.0 M NaCl in 0.25 M sodium phosphate (pH 6.0) at a flow rate of 1 mL/min. Inhibitory fractions were separately applied to a Brownlee C8 RP-HPLC column and eluted with a gradient of 0%–100% acetonitrile containing 0.1% trifluoroacetic acid (TFA) at a flow rate of 0.3 mL/min. Trypsin inhibition was routinely monitored using *para*-nitroanilide (pNA) tetrapeptide substrates through the purification process. The final fractions containing purified inhibitors were subjected to MALDI/TOF-MS (Bio-Rad, Sydney, Australia) for analysis of homogeneity and masses.

### 4.3. Amino-Acid Sequence Determination Using LC–MS/MS

Each inhibitor was equilibrated in 50 mM ammonium bicarbonate; then, the disulfide bonds were reduced by incubation with 5 mM 1,4-dithiothreitol (95 °C, 5 min) and cysteines were subsequently alkylated by addition of 10 mM iodoacetamide (RT, 20 min, in the dark). Then the inhibitor was digested with sequencing-grade modified trypsin (Thermo Fisher Scientific Pierce) using an enzyme-to-substrate ratio of 1:40 (*w*/*w*). Digestion proceeded at 37 °C for 2 h, and then at 30 °C overnight. Digestion was terminated by adding 1% TFA to the mixture.

Digestion mixtures were subjected to LC–MS/MS on a QTRAP 4500 mass spectrometer (SCIEX) integrated with a Shimadzu LC system and controlled by Analyst 1.6.2 Software (SCIEX, Singapore). The digestion mixture was loaded into a Brownlee C8 RP-HPLC column (2.1 × 30 mm) and eluted with a linear gradient of 5%–70% acetonitrile containing 0.1% formic acid at a flow rate of 0.2 mL/min. The eluates were ionized using the electrospray ionization Turbo Spray source in positive mode. The mass spectrometer was operated in information-dependent acquisition (IDA) mode, automatically switching between a survey MS scan (*m*/*z* 300–2000) and three MS/MS scans (*m*/*z* 50–1800). 

MS/MS data (WIFF format) were converted to MGF format by an MS data converter (SCIEX, Singapore) and visualized using the seeMS program (ProteoWizard, CA, USA) [49,50]. Unambiguous peptide sequence fragments were deduced from spacing between adjacent product ions of the same series in the MS/MS spectra. These sequence fragments were searched against a whole-body transcriptome of *A. tenebrosa* to identify candidate sequences. The theoretical digested peptide fragments of these sequences were compared with the monoisotopic peaks extracted from the MS spectra to enable reconstruction of the complete sequences of the inhibitors. 

### 4.4. Transcriptomic Analysis

Total RNA was extracted from a single whole specimen of *A. tenebrosa* that was homogenized in liquid nitrogen [42,51]. RNA was visually inspected on a 1.5% agarose gel stained with GelRed (Biotium). Further quality assessment was performed using an RNA nano chip on a Bioanalyzer 2100 (Agilent) to confirm RNA integrity and concentration. RNA libraries were prepared using the TruSeq Stranded mRNA Library Preparation Kit (Illumina, San Diego, CA, USA) as described in the manufacturer’s instructions. Resulting complementary DNA (cDNA) quality and concentration were assessed using a Bioanalyzer 2100 (Agilent, Santa Clara, CA, USA) with a high-sensitivity DNA chip. One-hundred and fifty base pair (bp) paired-end libraries were sequenced on an Illumina NextSeq 500 instrument (Illumina, San Diego, CA, USA). 

High-quality reads (>Q30, <1% ambiguities) were assembled using the Trinity de novo assembler software version 2.2.0 (Broad Institute, Cambridge, MA, USA) [52]. Default settings, as well as the Trimmomatic flag, were used to assemble contiguous sequences and remove any low-quality reads and adaptors [52]. Differential expression of ATPI-I across tentacles, mesenterial filaments, and acrorhagi tissues was determined using the samples and methods of Surm et al. [42].

### 4.5. MALDI Mass Spectrometry Imaging

MALDI-MS imaging (MSI) was guided by published protocols [53,54] but with sample preparation optimized as recently described [55,56,57]. Briefly, specimens of *A. tenebrosa* were left in 50% RCL2 (Alphelys, France) fixative/ethanol at room temperature overnight, sequentially dehydrated into ethanol (3 × 15 min at each concentration), cleared in xylene for 30 min, and embedded in paraffin wax. A whole embedded animal was sectioned transversally at 7 µm thickness. Sections were de-paraffinized by careful washing with xylene, and optically imaged prior to applying α-cyano-4-hydroxycinnamic acid (7 mg/mL in 50% acetonitrile, 0.2% TFA) using a Bruker ImagePrep automated matrix sprayer. FlexControl 3.3 (Bruker, Germany) was used to operate an UltraFlex III TOF/TOF mass spectrometer (Bruker, Germany) in linear positive mode, with the range set to 1000–20,000 *m/z*. A small laser size was chosen to achieve a spatial resolution of 50 µm, and matrix ion suppression was enabled up to 980 *m/z*. Individual MSI experiments were performed using FlexImaging 4.0 (Bruker, Germany). FlexImaging was used to establish the geometry and location of the section on the slide based upon the optical image, choose the spatial resolution, and call upon FlexControl to acquire individual spectra, accumulating 200 shots per raster point. FlexImaging was subsequently used to visualize the data in two-dimensional (2D) ion-intensity maps, producing an averaged spectrum based upon the normalized individual spectra collected during the experiment. Following MSI data acquisition, the section was stained using regular hematoxylin and eosin protocol as described elsewhere [58].

### 4.6. Synthesis of Peptide para-Nitroaniline (pNA) Substrates

The pNA substrates were synthesized on 2-chlorotrityl chloride resin (1.55 mmol Cl^−^/g, Iris Biotech), following an adjusted solid-phase peptide synthesis protocol for Fmoc chemistry [59]. Detailed methods are described in the Appendix A.

### 4.7. Recombinant Production and Purification of KLKs

Pro-KLK5 and pro-KLK7 were expressed in the yeast *Pichia pastoris* strain X-33 as previously described [15,60,61]. Pro-KLK14 was expressed in stably transfected Sf9 insect cells (*Spodoptera frugiperda*) following previously published protocols. Pro-KLKs were enriched, activated, and purified as previously described [40]. Active-site concentrations were titrated using known covalent inhibitors which bind to the protease with 1:1 stoichiometry. Detailed methods are described in the Appendix A. 

### 4.8. Kinetic Assays

The inhibitory properties of inhibitors against trypsin (Sigma T8003), chymotrypsin (Sigma C4129), KLK5, KLK7, and KLK14 were determined using a fixed concentration of protease pre-incubated with serial dilutions of the inhibitor in assay buffer (0.1 M Tris-Cl (pH 8.0) containing 0.1 M NaCl, 0.005% Triton X-100, and 0.05% NaN_3_) at room temperature for 20 min to reach equilibrium. Assays were initiated by addition of a fixed amount of respective chromogenic substrates: *N-*acetyl-Tyr-Ala-Ser-Arg-*para*-nitroanilide (Ac-YASR-pNA) for trypsin, KLK5, and KLK14, *N*-acetyl-Gly-Arg-Pro-Tyr-*para*-nitroanilide (Ac-GRPY-pNA) for chymotrypsin, and Lys-His-Leu-Tyr-*para*-nitroanilide (KHLY-pNA) for KLK7. Substrate hydrolysis rates were determined by measuring the change in absorbance at 405 nm (ΔOD_405 nm/min_) for 5 min using a Benchmark Plus microplate spectrophotometer (Bio-Rad, Sydney, Australia). The degree of inhibition at each inhibitor concentration was normalized as a percentage of residual protease activity compared to uninhibited protease activity. Inhibition constants (*K*_i_) were determined using the Morrison equation for tight-binding inhibition [62] in GraphPad Prism 5.01 (San Diego, CA, USA). 

### 4.9. Model Building and Molecular Dynamics Simulations

The full sequences of ATPI-I, ATPI-II, and KLK14 (UniProt: Q9P0G3) were separately subjected to homology modeling in YASARA (Yet Another Scientific Artificial Reality Application, version 17.8.15 Vienna, Austria) [34]. The complexes of *A. tenebrosa* inhibitors and KLKs were built in YASARA using the above models and the following crystal structures in the Protein Data Bank (PDB): KLK5 (PDB: 2PSX) and KLK7 (PDB: 2QXI). Each complex was solvated in a TIP3P (transferable intermolecular potential with 3 points) water box, neutralized with 0.1 M Na^+^ and Cl^−^ counter ions, and parameterized with the CHARMM27 force field [63] using VMD (Visual Molecular Dynamics) version 1.9.2 (Urbana-Champaign, Il, USA) [64]. After energy minimization and step-wise relaxation, all systems were simulated under NPT (constant number of atoms, potential and temperature) ensembles for 20 ns using NAMD (Nanoscale Molecular Dynamics) version 2.9 (Urbana-Champaign, Il, USA) [65]. Full details are described in the Appendix A.

### 4.10. Sequence Consensus Analysis

The previously identified sequences of ATPI-I and ATPI-II were used to search for Kunitz domain homologues in the transcriptomic library derived from *A. tenebrosa* established by Prentis et al. [26]. Searches were conducted using BLASTN and BLASTP algorithms. Sequence analysis was performed using the Jalview suite of programs [66]. Conservation scores based on residue physico-chemical properties were calculated at each position using the Jalview built-in AMAS (analysis of multiple aligned sequences) algorithm [66,67]. This algorithm records the residues occurring at a position and analyzes their properties (hydrophobic, positive, negative, polar, charged, small, tiny, aliphatic, aromatic, proline, and their negation), resulting in a score ranging from 0 (unconserved across all sequences) to 11 (identical across all sequences) [67]. 

### 4.11. In Silico Alanine Scanning

Alanine scanning mutagenesis is an energy-based method widely used to identify residues that are important in maintaining thermostability of the protein or complexes [68,69]. In this study, alanine scanning experiments were performed using the FoldX plugin (version 3.0b4, Barcelona, Spain) [70] in YASARA 17.8.15 [34] to study the energy decomposition of each residues in *A. tenebrosa* inhibitors. Firstly, all Kunitz domain sequences identified in the *A. tenebrosa* transcriptomic library were subjected to homology model construction and minimization in YASARA [34]. Then, inhibitor structures were subjected to the “RepairPDB” command to fix side chains with bad torsion angles, van der Waals clashes, or “out-of-range” energies. Alanine scanning was run using the FoldX built-in position-specific substitution matrix (PSSM) method to calculate the change in free energy (ΔΔG) after mutating each of the selected residues to alanine: ΔΔG = ΔG_mut_ − ΔG_wt_, where ΔG_mut_ and ΔG_wt_ refer to the free energy of unfolding for mutated protein and wild-type protein, respectively [70].

## Figures and Tables

**Figure 1 marinedrugs-17-00701-f001:**
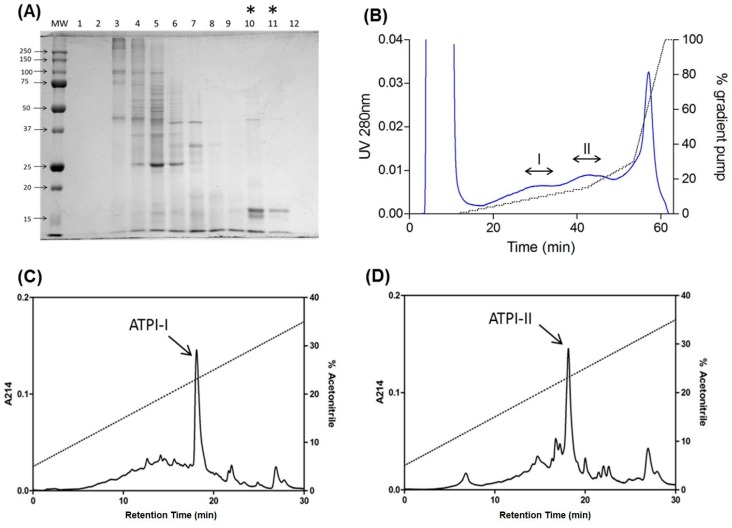
Purification of two protease inhibitors, ATPI-I and ATPI-II, from *Actinia tenebrosa*. (**A**) SDS-PAGE gel visualization of fractions eluted from gel filtration using Sephadex S-200 resin. Fractions 1–12 were collected according to elution time. Inhibitory fractions are indicated by asterisks. (**B**) UNOsphere S cation exchange chromatography revealed two distinct inhibitory fractions, labeled “I” and “II”; (**C**) reverse-phase (RP)-HPLC purification of inhibitor ATPI-I; (**D**) RP-HPLC purification of inhibitor ATPI-II.

**Figure 2 marinedrugs-17-00701-f002:**
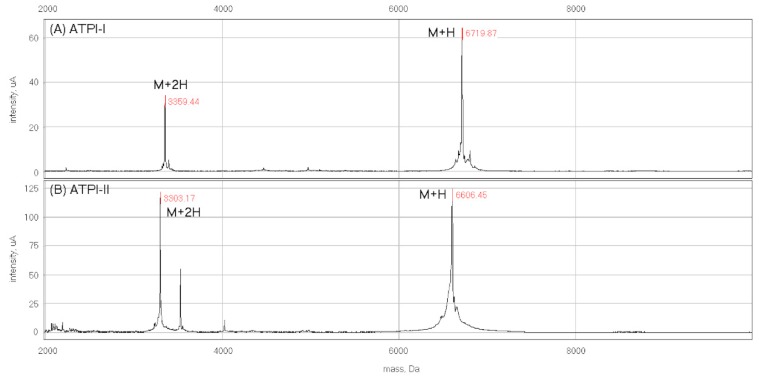
The masses and homogeneity of ATPI-I and ATPI-II. Purified *A. tenebrosa* inhibitors (**A**) ATPI-I and (**B**) ATPI-II were analyzed by matrix-assisted laser desorption/ionization time-of-flight mass spectrometry (MALDI/TOF-MS). The MS spectrum of ATPI-I showed a singly charged ion of 6719.9 Da and a doubly charged ion of 3359.4 Da, while the MS spectrum of ATPI-II showed a singly charged ion of 6606.5 Da and a doubly charged ion of 3303.2 Da.

**Figure 3 marinedrugs-17-00701-f003:**
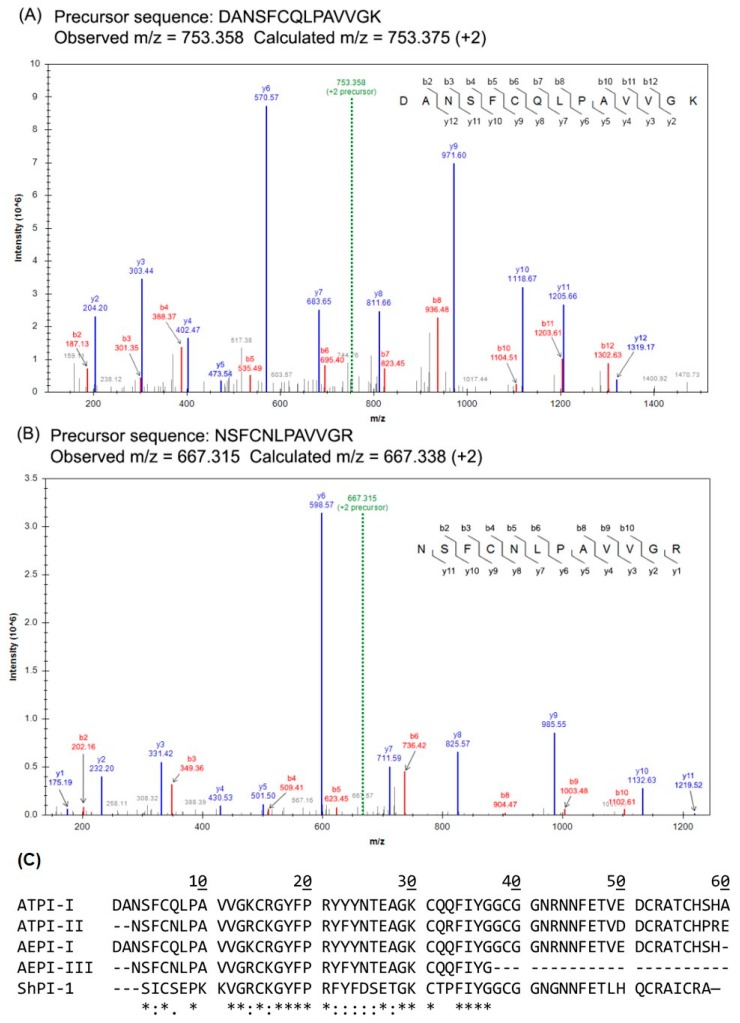
Representative MS/MS spectra for ATPI-I and ATPI-II. ATPI-I and ATPI-II were digested with trypsin and analyzed by LC–MS/MS. (**A**) ATPI-I N-terminal sequence _1_DANSFCQLPAVVGK_14_ and (**B**) ATPI-II N-terminal sequence _3_NSFCNLPAVVGR_14_ were deduced from MS/MS spectra. Observed *m*/*z* values of the precursor ions are consistent with calculated values. Peaks corresponding to theoretical product ions are labeled. Most peaks were identified to be *b*- and *y*-series ions, as well as a few *a-* and *z-*series ions. All cysteines were alkylated by iodoacetamide, resulting in a +57.02 Da mass shift for this residue. (**C**) Full sequences of ATPI-I and ATPI-II in comparison with AEPI-I and AEPI-II from *A. equina* [27], and ShPI-1 from *Stichodactyla helianthus* [28]. Multiple alignment was performed using Clustal Omega [29]. An asterisk (*) indicates a fully conserved residue; a colon (:) indicates conservation between residues of strongly similar properties; a period (.) indicates conservation between residues of weakly similar properties [29].

**Figure 4 marinedrugs-17-00701-f004:**
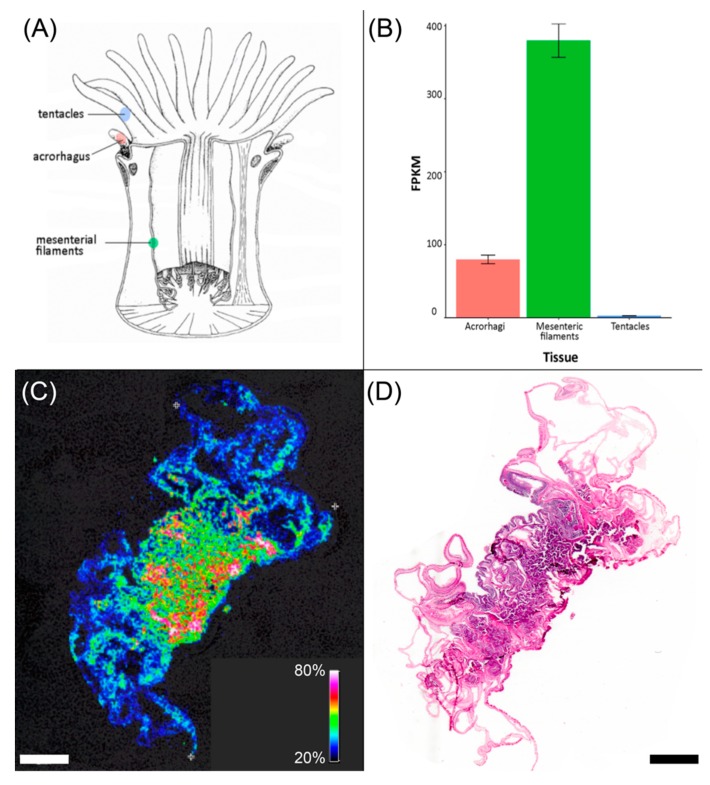
Tissue distribution of ATPI-I in *A. tenebrosa*. (**A**) Anatomy of a generalized sea anemone (adapted from Reference [33]), showing the locations of tentacles, acrorhagus, and mesenterial filaments. (**B**) Tissue distribution of messenger RNA (mRNA) of ATPI-I in *A. tenebrosa*. ATPI-I mRNA is expressed highly in the mesenteric filaments (green) and also present in acrorhagi (red) and to a lesser extent in tentacles (green). FPKM: fragments per kilobase of transcript per million mapped reads. (**C**) MALDI-MS imaging color map showing the distribution of ATPI-I across a section of *A. tenebrosa*, which was (**D**) subsequently stained with hematoxylin and eosin. Color in (**C**) corresponds to the signal intensity, and scale bars represent 2 mm.

**Figure 5 marinedrugs-17-00701-f005:**
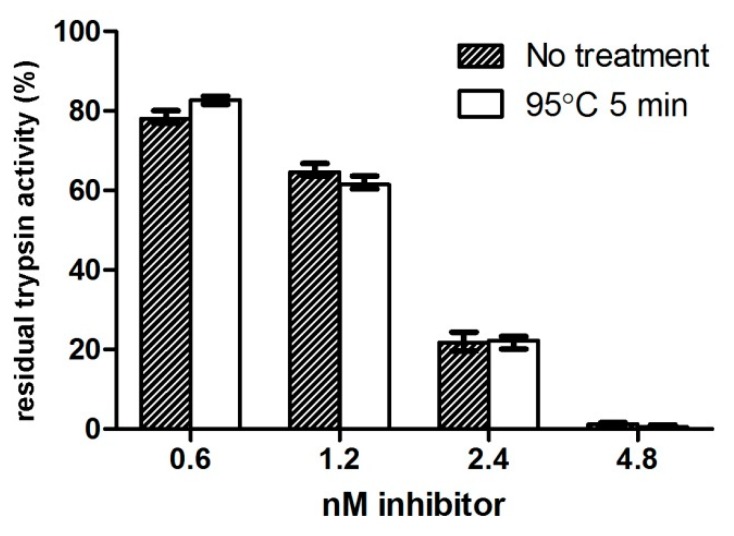
ATPI-I shows no loss of inhibitory activity after heat treatment. ATPI-I solution was incubated at 95 °C for 5 min and assayed against 3 nM of trypsin using Ac-YASR-pNA (*para-*nitroaniline) as substrate (*n* = 3). Trypsin catalytic rate (ΔOD_405 nm_) was measured after incubation with four concentrations of untreated and heat-treated inhibitor and is presented as a percentage (%) of the catalytic rate of uninhibited trypsin. ATPI-I showed no loss of inhibitory activity following heat treatment.

**Figure 6 marinedrugs-17-00701-f006:**
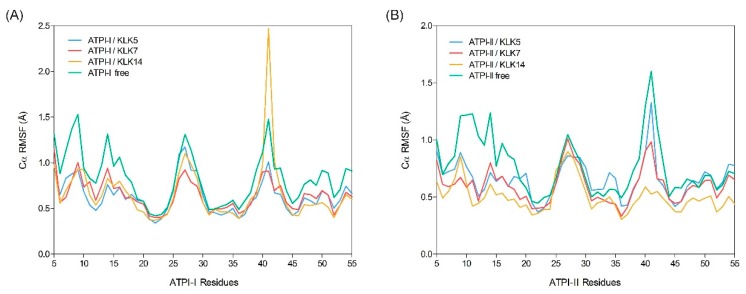
Root-mean-square fluctuation (RMSF) of Cα atoms of *A. tenebrosa* inhibitors during molecular dynamics simulation. *A. tenebrosa* inhibitors (**A**) ATPI-I and (**B**) ATPI-II were simulated in free form (green line) and in complex with kallikrein-related peptidase 5 (KLK5) (blue line), KLK7 (red line), and KLK14 (orange line). Cα RMSF values were averaged over the last 5 ns of the trajectories (*n* = 3). Residues at the N- and C-terminus are not shown.

**Figure 7 marinedrugs-17-00701-f007:**
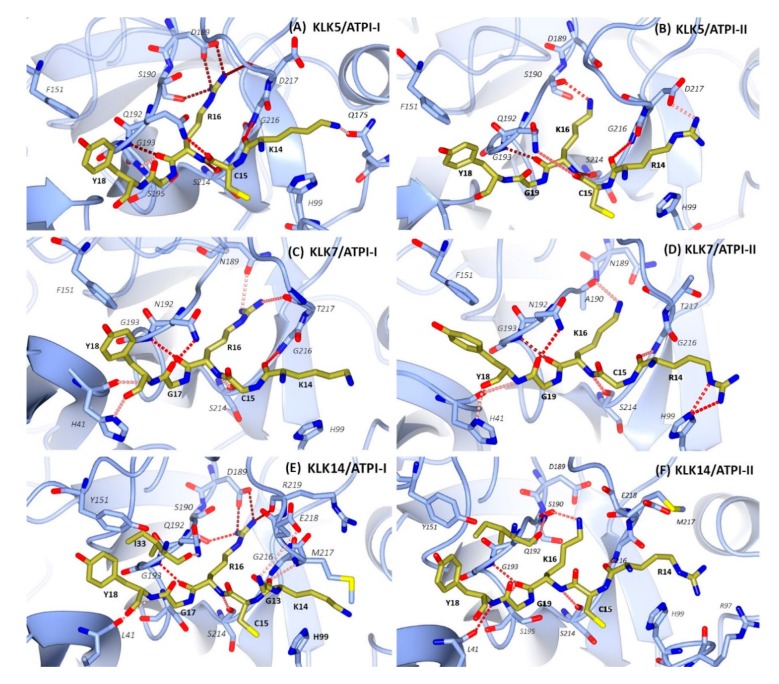
Molecular modeling of ATPI inhibitors in complex with selected KLKs. The average structures were calculated from molecular dynamics simulations of the following complexes: (**A**) KLK5/ATPI-I, (**B**) KLK5/ATPI-II, (**C**) KLK7/ATPI-I, (**D**) KLK7/ATPI-II, (**E**) KLK14/ATPI-I, and (**F**) KLK14/ATPI-II. Proteases are depicted by blue ribbons, with the contacting residues shown in stick mode and carbon atoms colored blue. The contact residues of inhibitors (shown in stick mode with carbon atoms colored gold) lie in the active site of the protease. Intermolecular hydrogen bonds are depicted with dashed lines. Hydrogen bond occupancy is indicated by the shades of red, from pale pink (30%) to tan (>85%).

**Figure 8 marinedrugs-17-00701-f008:**
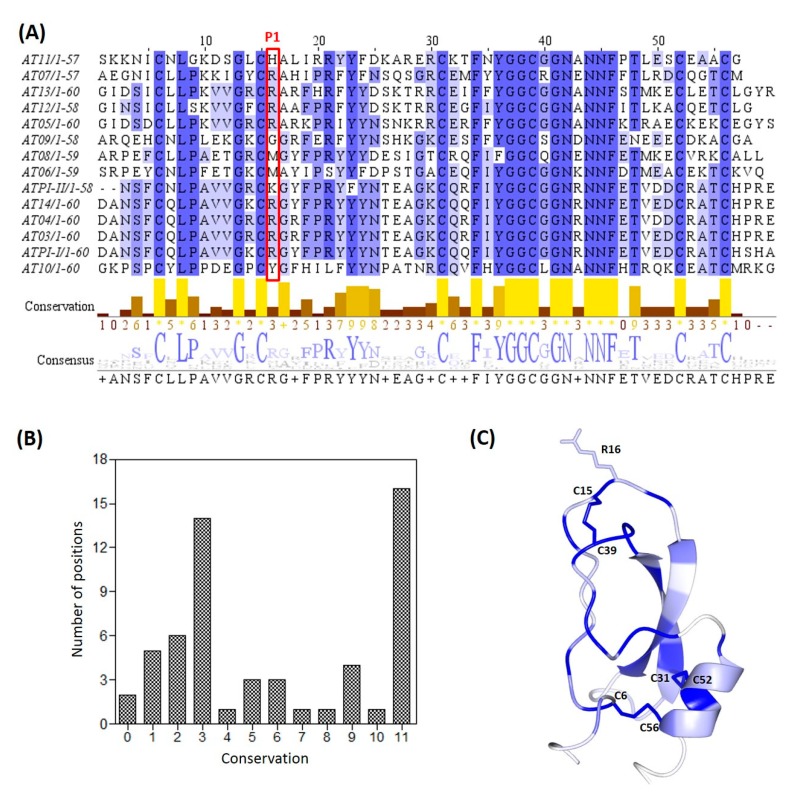
Multiple sequence alignment of 14 Kunitz homologs identified in an *A. tenebrosa* whole-body transcriptome. (**A**) The sequences of ATPI-I and ATPI-II were aligned with 12 deduced sequences (A03–A14) of Kunitz homologs found in an *A. tenebrosa* transcriptomic library. Conservation scores were calculated based on physico-chemical properties. The P1 residue is indicated by a red rectangle. (**B**) The number of positions belonging to each conservation score, which ranges from 0 to 11. (**C**) Molecular model of ATPI-I is shown in ribbon form and colored by conservation scores from 0 (white) to 11 (blue). The P1 residue Arg16 and six conserved Cys residues are shown in stick mode, with numbering according to their position in ATPI-I.

**Figure 9 marinedrugs-17-00701-f009:**
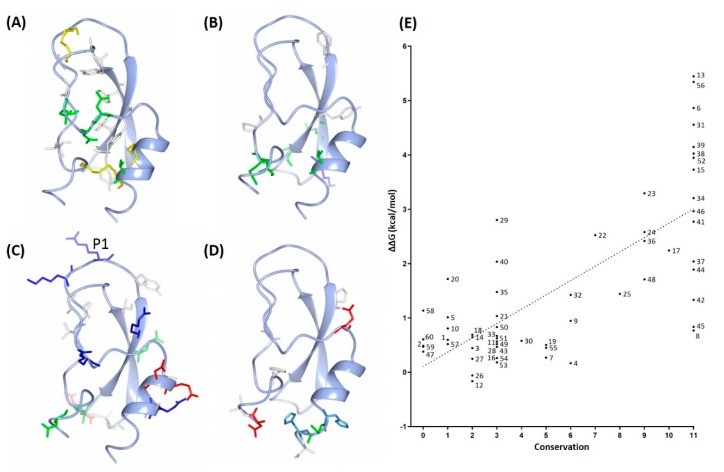
Predicted model of Kunitz domain from *A. tenebrosa* and the calculated per-residue contribution to stability (ΔΔG). (A–D) Predicted model of Kunitz domain (shown in ribbon mode) from *A. tenebrosa*, highlighting residues (shown in stick mode) with conservation level scores of (**A**) 9–11, (**B**) 5–8, (**C**) 2–4, and (**D**) 0–1. The highlighted residues are colored according to residue properties: non-polar residues are colored white, polar uncharged residues are colored green, basic residues are colored blue, and acidic residues are colored red. (**E**) The calculated per-residue contribution to stability (ΔΔG) was plotted against degree of conservation for each position.

**Table 1 marinedrugs-17-00701-t001:** Sequence analysis of two inhibitors isolated from *Actinia tenebrosa*.

Inhibitor	Amino-acid Sequence	Mass/Charge (Charge)	
		Observed	Calculated
ATPI-I	_1_DANSFCQLPAVVGK_14_	753.47 (+2)	753.37 (+2)
	_15_CRGYFPRYYYNTEAGK_30_	510.32 (+4)	509.97 (+4)
	_22_YYYNTEAGK_30_	554.97 (+2)	554.75 (+2)
	_31_CQQFIYGGCGGNR_43_	758.89 (+2)	758.82 (+2)
	_31_CQQFIYGGCGGNRNNFETVEDCR_53_	928.00 (+3)	927.72 (+3)
	_44_NNFETVEDCR_53_	642.39 (+2)	642.27 (+2)
	_44_NNFETVEDCRATCHSHA_60_	513.00 (+4)	512.71 (+4)
ATPI-II	_3_NSFCNLPAVVGRCKGYFPR_21_	561.60 (+4)	561.28 (+4)
	_3_NSFCNLPAVVGR_15_	667.53 (+2)	667.34 (+2)
	_15_CKGYFPR_21_	464.63 (+2)	464.23 (+2)
	_22_YFYNTEAGKCQR_33_	513.23 (+3)	512.90 (+3)
	_34_FIYGGCGGNRNNFETVDDCRATCHPRE_60_	641.54 (+5)	641.28 (+5)

**Table 2 marinedrugs-17-00701-t002:** Inhibition constants (*K*_i_) for the inhibitors ATPI-I and ATPI-II against a spectrum of serine proteases. KLK—kallikrein-related peptidase; pNA—*para-*nitroaniline.

Protease	Substrate	*K*_i_ (nM)	
		ATPI-I	ATPI-II
Trypsin	Ac-YASR-pNA	0.050 ± 0.007	0.080 ± 0.008
Chymotrypsin	Ac-GRPY-pNA	7.4 ± 1.2	2.9 ± 0.3
KLK5	Ac-YASR-pNA	1.6 ± 0.2	21 ± 1
KLK7	KHLY-pNA	1.4 ± 0.1	3.2 ± 0.2
KLK14	Ac-YASR-pNA	13 ± 1	89 ± 7

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
