# Peer review of "A Versatile and Robust Serine Protease Inhibitor Scaffold from Actinia tenebrosa"

_marinedrugs, 2019, doi:10.3390/md17120701_

Round 1

Reviewer 1 Report

Xingchen Chen et al. have carried out an extensive and rigorous study of Kunitz type peptides of sea anemone Actinia tenebrosa, they isolated and identified the sequences of two new peptides from the whole anemone body through engaging both classical methods of protein chemistry and tandem mass spectrometry methodologies. It is noteworthy that these inhibitors share both high sequence similarity and fold, but are able to discriminate against closely related proteases. MALDI-MSI provided us with new facts regarding the tissue distribution of ATPI-I in A. tenebrosa that contribute to long-debated issue about the biological role of Kunitz inhibitors in sea anemones. Moreover, the analysis of the A. tenebrosa transcriptome library revealed an additional 12 sequences of peptides, which are supposed to adapt this fold. This evidences to the presence of a Kunitz peptides combinatorial library in A. tenebrosa, similar to other sea anemones, for example, Heteractis crispa. Then, biophysical and in silico (including homology modelling and molecular dynamic simulations as well as an alanine scanning mutagenesis) approaches have been used to characterize and elucidate the individual residue contributions for the peptide’s stability, respectively. All these data support the main conclusion.

To strengthen the conclusion however, following points should be addressed

In my opinion, the expression "a growing focus" is not very felicitous (line 74) Apparently instead of «Kinetic assays were performed to determined inhibition constants…» the authors had in mind «Kinetic assays were performed to determine inhibition constants…» (line 175) It would probably be useful to add the sequences of AEPI-I and N-terminal fragment of AEPI-III (inhibitors isolated from equine), as well as ShPI isolated from S. helianthus in Figure 3(C), so that it would be more visual. Moreover, in the caption to the figure should be added a decoding of characters «», «» and «». Figure 8 (B) seems redundant because the diagram repeats the data shown in Figure 8 (A). Two points at the end of the sentence (line 292). The conservation scores mapping on the model of ATPI-I structure is very useful for understanding the significance of various structural elements in the molecule stability as well as in its functionality. However, the representation in the form of a blue gradient does not allow one to discriminate unequivocally the color differences derived from conservativeness from those related to just the volumetric image of the structure (compared for example to figure 7). It remains not clear enough what exactly the authors evaluated during the in silico alanine mutagenesis: was it the inhibitor molecule stability (lines 313-315 «In order to estimate the per-residue contribution to molecule stability, in silico alanine scanning mutagenesis was performed by calculation of the change in free energy (ΔΔG) after mutating each of the residues to alanine…») and/or the stability of the protease/inhibitor complex (line 318). The authors proposed «residues with ΔΔG > 2 kcal/mol after mutation to alanine are considered significant in the maintenance of the stability of the protease/inhibitor complex» However, this is too much to say based on the data demonstrated in Results and Materials and Methods Please, add a color scheme for side chains (into the caption) or residue numbers to the figures 9(A-D). Kunitz protease inhibitors interaction with their targets is studied broadly and intensively by various experimental and computational methods, but this phenomenon conceals quite a lot to be penetrated still. PDB data bank contains structures of various protease/inhibitor complexes. The authors succeed in finding of interdependence between the probable interactions of the components in modeled complexes and the peptides inhibitory potency. However, the use of simply homology docking to construct the structure models of the protease/inhibitor complexes with the ShPI-1/trypsin (PDB: 3T62) structure as a template (Supplementary Methods) does not seem justified. Taking into account the presence of charged residues substitutions, the model construction for the enzyme/peptide complex (for each case) should be carried out more carefully using homologous docking (if any) in combination with other docking procedures or involving data derived from experiment, mutagenesis for instance.

Author Response

We thank the reviewers for their consideration of our manuscript and respond to their comments as follows

Reviewer 1’s comments and responses:

“In my opinion, the expression "a growing focus" is not very felicitous (line 74)”

Response: this has been changed to “a growing interest”.

“Apparently instead of «Kinetic assays were performed to determined inhibition constants…» the authors had in mind «Kinetic assays were performed to determine inhibition constants…» (line 175)”

Response: this has been corrected (section 2.4 line 181 after revision).

“It would probably be useful to add the sequences of AEPI-I and N-terminal fragment of AEPI-III (inhibitors isolated from A. equina), as well as ShPI isolated from  helianthusin Figure 3(C), so that it would be more visual. Moreover, in the caption to the figure should be added a decoding of characters «*», «.» and «:».”

Response: Figure 3 has been replaced. The sequences of AEPI-I, AEPI-III and ShPI-1 have now been added to Figure 3C and an explanatory note added regarding conservation indicators.

“Figure 8 (B) seems redundant because the diagram repeats the data shown in Figure 8 (A).”

Response: the two panels show different information and focus on different aspects of sequence analysis. Figure 8A presents the sequences of all homologs and the conservation scores of each position. Figure 8B counts the number of positions at each conservation score, so we can have a direct view of the frequency of each conservation score. In Figure 8B, the y-axis label has been changed to “Number of positions” instead of “Number of residues” to give a more accurate description of the data.

“Two points at the end of the sentence (line 292).”

Response: the extra point has been deleted.

“The conservation scores mapping on the model of ATPI-I structure is very useful for understanding the significance of various structural elements in the molecule stability as well as in its functionality. However, the representation in the form of a blue gradient does not allow one to discriminate unequivocally the color differences derived from conservativeness from those related to just the volumetric image of the structure (compared for example to figure 7).”

Response: Figure 8 has been replaced. Now the structure in Figure 8C shows clear gradient from white to blue according to the conservation scores in Figure 8A.

“It remains not clear enough what exactly the authors evaluated during the in silico alanine mutagenesis: was it the inhibitor molecule stability (lines 313-315 «In order to estimate the per-residue contribution to molecule stability, in silico alanine scanning mutagenesis was performed by calculation of the change in free energy (ΔΔG) after mutating each of the residues to alanine…») and/or the stability of the protease/inhibitor complex (line 318). The authors proposed «residues with ΔΔG > 2 kcal/mol after mutation to alanine are considered significant in the maintenance of the stability of the protease/inhibitor complex» However, this is too much to say based on the data demonstrated in Results and Materials and Methods”

Response: A separate section (section 4.11) has been added to Materials and Methods to give details of the alanine scanning analysis. In short, alanine scanning is a method that can determine the contribution of a specific residue to protein stability or complex binding, and has been used in various research previously (see references 69 for a comprehensive review of this method). The formula for calculating ΔΔG is described in section 4.11, reflecting the relationship between ΔΔG and protein stability.

For all residues, an average ∆∆G value of 1.6 kcal/mol was obtained from alanine scanning. Hence, we previously used a cutoff of 2 kcal/mol, which is a more stringent value, to determine whether a residue is important for protein stability. Now the sentence has been changed to give the actual range and average value of ∆∆G.

“Please, add a color scheme for side chains (into the caption) or residue numbers to the figures 9 (A-D).”

Response: colour scheme of sidechains has been added into the caption.

“Kunitz protease inhibitors interaction with their targets is studied broadly and intensively by various experimental and computational methods, but this phenomenon conceals quite a lot to be penetrated still. PDB data bank contains structures of various protease/inhibitor complexes. The authors succeed in finding of interdependence between the probable interactions of the components in modeled complexes and the peptides inhibitory potency. However, the use of simply homology docking to construct the structure models of the protease/inhibitor complexes with the ShPI-1/trypsin (PDB: 3T62) structure as a template (Supplementary Methods) does not seem justified. Taking into account the presence of charged residues substitutions, the model construction for the enzyme/peptide complex (for each case) should be carried out more carefully using homologous docking (if any) in combination with other docking procedures or involving data derived from experiment, mutagenesis for instance.”

Response: Homology docking is only used to construct the initial structure of the protease-inhibitor complex. Since Kunitz inhibitors belong to the standard mechanism inhibitor superfamily, they bind to serine proteases in a similar substrate-like manner.  So we used the crystal structure of trypsin-ShPI to guide the initial position of binding. Then, in order to obtain a stable complex, we subjected these initial complexes to solvation, energy minimization, equilibrium run with step-wise relaxation of sidechain and mainchain atoms, followed by three independent simulations (described in supplementary methods). The effect of charged residue substitutions was taken into account in these steps. After the system reached equilibrium, the simulation trajectories were extracted and used for analysis of H-bonds, close contacts, etc.

Reviewer 2 Report

The authors have written a very clear manuscript describing the diversity and structural attributes of these serine protease inhibitors from a cnidarian.  This research will surely be of interest to this field of researchers as well as a broader audience. I have a few minor edits and suggestions for the authors to consider in a revision. These edits are primarily in description of the methods, which I think will clarify some areas where additional clarity may be useful or where I was uncertain why particular decisions were made.  

Number of individuals used for transcriptome sequencing. The authors should provide the number of anemones used in the extraction of RNA (sections 4.1 and 4.2). Because multiple individuals appear to have been used, is it possible the authors have detected allele variants and not independent genes? On line 440 the authors refer to anemones as belonging to a "lower order". This should be corrected because this comments suggests a hierarchical relationship of animals, which is clearly not accurate. All extant species have had the same time to evolve. Section 4.10. Could the authors provide some additional details on the construction of the phylogeny and justification for use of BLOSUM62? Figure 4. Panel C and D would be more interpretable with labels of what parts of the anemone are present in this cross-section. Figure 9E. The points in this plot are difficult to see, can the authors increase the font size?  Also, would it be useful to add a regression analysis of these data to indicate the fit of their model?

Author Response

We thank the reviewers for their consideration of our manuscript and respond to their comments as follows

Reviewer 2’s comments and responses:

“Number of individuals used for transcriptome sequencing. The authors should provide the number of anemones used in the extraction of RNA (sections 4.1 and 4.2). Because multiple individuals appear to have been used, is it possible the authors have detected allele variants and not independent genes?”

Response: RNA extraction was performed using a single specimen. This has now been described in section 4.4. For inhibitor characterization (amino acid sequence analysis and kinetic assays), a considerable amount of inhibitor is required, so multiple individuals were used in order to extract adequate protein materials. The homogeneity of purified inhibitors was validated by mass spectrometry.

“On line 440 the authors refer to anemones as belonging to a "lower order". This should be corrected because this comments suggests a hierarchical relationship of animals, which is clearly not accurate. All extant species have had the same time to evolve.”

Response: This sentence now reads “Because sea anemones have a simple nervous system, use of A. tenebrosa does not require animal ethics approval.” (section 4.1 line 449-450 in the revised manuscript).

“Section 4.10. Could the authors provide some additional details on the construction of the phylogeny and justification for use of BLOSUM62?”

Response: Section 4.10 has been revised to give more accurate information. For conservation scoring, we used the Jalview built-in AMAS algorithm which performed a direct comparison based on physico-chemical properties of amino acids shared at a position without generating a tree (see reference [67] for more details of this algorithm).

We used this method because it is flexible, allowing each residue property to be examined independently, and it reflects the tolerance of different residues at a position, which can be used as a guide in future inhibitor engineering. This was mentioned in lines 364-367 and lines 440-443. Additionally, the analysis was carried out by direct comparison of aligned residues without constructing a phylogenetic tree.

“Figure 4. Panel C and D would be more interpretable with labels of what parts of the anemone are present in this cross-section.”

Response: Panel A is included as a guide to the identity of tissues in the other panels.

“Figure 9E. The points in this plot are difficult to see, can the authors increase the font size?  Also, would it be useful to add a regression analysis of these data to indicate the fit of their model?” 

Response: Figure 9 has been replaced with new figure and the font size in Figure 9E has been increased. Linear regression analysis has been described in the text (line 362).

Round 2

Reviewer 1 Report

The authors have made the appropriate additions and corrections, and significantly improved the manuscript .

Regarding the homology docking:

I have built homology models of ATPI-I and ATPI-II and found that variable Q33R is buried and not exposed to solvent. And the molecular surfaces of ATPI-I and prototype fit well enough at the region of canonical binding loop. This is a special case for which the docking procedure according to homology can be applied. Just a small remark. This is not a general rule. When starting state for MD is generated, the use of homology docking alone will not be justified for the construction of complexes for enzymes with other inhibitors (found from a transcriptome, for example) and especially for cases when there is a charge inversion during the residues substitution.

      G M T    
Определить язык Азербайджанский Албанский Английский Арабский Армянский Африкаанс Баскский Белорусский Бенгальский Бирманский Болгарский Боснийский Валлийский Венгерский Вьетнамский Галисийский Греческий Грузинский Гуджарати Датский Зулу Иврит Игбо Идиш Индонезийский Ирландский Исландский Испанский Итальянский Йоруба Казахский Каннада Каталанский Китайский (Упр) Китайский (Трад) Корейский Креольский (Гаити) Кхмерский Лаосский Латинский Латышский Литовский Македонский Малагасийский Малайский Малайялам Мальтийский Маори Маратхи Монгольский Немецкий Непали Нидерландский Норвежский Панджаби Персидский Польский Португальский Румынский Русский Себуанский Сербский Сесото Сингальский Словацкий Словенский Сомали Суахили Суданский Тагальский Таджикский Тайский Тамильский Телугу Турецкий Узбекский Украинский Урду Финский Французский Хауса Хинди Хмонг Хорватский Чева Чешский Шведский Эсперанто Эстонский Яванский Японский   Азербайджанский Албанский Английский Арабский Армянский Африкаанс Баскский Белорусский Бенгальский Бирманский Болгарский Боснийский Валлийский Венгерский Вьетнамский Галисийский Греческий Грузинский Гуджарати Датский Зулу Иврит Игбо Идиш Индонезийский Ирландский Исландский Испанский Итальянский Йоруба Казахский Каннада Каталанский Китайский (Упр) Китайский (Трад) Корейский Креольский (Гаити) Кхмерский Лаосский Латинский Латышский Литовский Македонский Малагасийский Малайский Малайялам Мальтийский Маори Маратхи Монгольский Немецкий Непали Нидерландский Норвежский Панджаби Персидский Польский Португальский Румынский Русский Себуанский Сербский Сесото Сингальский Словацкий Словенский Сомали Суахили Суданский Тагальский Таджикский Тайский Тамильский Телугу Турецкий Узбекский Украинский Урду Финский Французский Хауса Хинди Хмонг Хорватский Чева Чешский Шведский Эсперанто Эстонский Яванский Японский            
      Звуковая функция ограничена 200 символами  
  Настройки : История : Обратная связь : Donate Закрыть

Author Response

We thank the reviewer for his diligence in modelling the two inhibitor complexes and for his commentary on the applicability of homology modelling as a starting point for analysing docked pairs. We will definitely bear his comments in mind when examining other instances of complex formation.